# Assessment of Pharmacists’ Knowledge and Practices towards Prescribed Medications for Dialysis Patients at a Tertiary Hospital in Riyadh Saudi Arabia

**DOI:** 10.3390/healthcare9091098

**Published:** 2021-08-25

**Authors:** Lolwa Al-Abdelmuhsin, Maha Al-Ammari, Salmeen D. Babelghaith, Syed Wajid, Abdulrahman Alwhaibi, Sultan M. Alghadeer, Mohamed N. Al Arifi, Ziyad Alrabiah

**Affiliations:** 1Pharmaceutical Care Services, Ministry of the National Guard—Health Affairs, Riyadh 11426, Saudi Arabia; l.s.alabdulmohsin@gmail.com; 2King Abdullah International Medical Research Centre, Riyadh 11481, Saudi Arabia; AmmariMa@ngha.med.sa; 3Clinical Pharmacy Department, College of Pharmacy, King Saud University, Riyadh 11451, Saudi Arabia; sbabelghaith@ksu.edu.sa (S.D.B.); aalwhaibi@KSU.EDU.SA (A.A.); salghadeer@ksu.edu.sa (S.M.A.); malarifi@ksu.edu.sa (M.N.A.A.); zalrabia@ksu.edu.sa (Z.A.)

**Keywords:** attitude, knowledge, hospital pharmacists, CRF, dialysis

## Abstract

Objective: The present study examined pharmacists’ knowledge and practices towards prescribed medications for hemodialysis patients. The impact of a pharmacist’s current positions and years of experience on practices and knowledge was also assessed. Methods: A cross-sectional survey was distributed to pharmacists working at King Abdul-Aziz Medical City-Central Region over a period of 4 months from July to October in 2015. Results: Of the 85 approached pharmacists, 66 pharmacists completed the questionnaire, among which 45 (68.2%), 9 (13.6%), and 12 (18.2%) of them were outpatient hospital pharmacists, discharge counselling pharmacists, and pharmacy practice residents, respectively. In total, 47 (55.3%) of the pharmacists sought drug information resources for newly prescribed medications to hemodialysis patients. Among the surveyed pharmacists, around two-thirds of them (63.6%) were completely confident during counselling hemodialysis patients, while 32% were moderately confident, and only 4.5% were not confident. All of the participating pharmacists checked each patient’s allergic status before dispensing hemodialysis medications. The majority of the outpatient hospital pharmacists (35; 77.8%), discharge pharmacists (8; 88.9%), and the pharmacy practice residents (11; 91.7%) agreed that oral ciprofloxacin should be given after dialysis session on the same dialysis days, while 18 (40%), 5 (55.6%), and 9 (75%) of the outpatient hospital pharmacists, discharge pharmacists, and pharmacy practice residents agreed that IV route is preferred for hemodialysis patients to administer epoetin alfa, respectively. Sixty-six percent of discharge pharmacists (*n* = 6), 91.7% (*n* = 11) of the pharmacy practice residents, and 55.6% (*n* = 25) of the outpatient hospital pharmacists checked patient laboratory results prior to dispensing medications (*p* = 0.001). Conclusions: Despite the limited knowledge regarding some prescribed medications, most of the hospital pharmacists showed good practices toward dialysis patients.

## 1. Introduction

Chronic renal failure (CRF) represents kidney damage from early- to end-stage renal disease (ESRD) and is a major public health condition worldwide [1]. The estimated global prevalence of patients having CRF ranges between 11% and 13%, while the global prevalence of patients with ESRD is 0.1% [2]. In Saudi Arabia, the total number of dialysis patients at the end of 2016 was reported to be 17,687, from which 16,315 patients were on hemodialysis and 1372 patients were on peritoneal dialysis [3]. The global prevalence of CRF patients has been increasing [4], and although more caution is usually followed upon taking care of these patients, stage 1 to 5 CRF patients, as well as hemodialysis patients, are still at a high risk for drug-related problems (DRPs) [4,5].

Pharmacists play an important role in maintaining patient safety. Many controlled trials have shown the impact of clinical pharmacists on general patient populations, by demonstrating that clinical pharmacists can reduce hospital admissions, length of hospital stay, readmissions, and emergency department visits [6,7,8]. A recent study among hemodialysis patients reported that patients were satisfied with the counseling provided by the pharmacist [9]. Another intervention-based randomized, controlled study found that pharmacists’ interventions were advantageous to patients with CRF, particularly in adhering to treatment guidelines [10]. Another recent review of pharmacists’ practices towards CRF patients found positive impacts of their interventions on the clinical, humanistic, and economic outcomes of CRF patients [11]. Moreover, multiple studies conducted to evaluate the impact of pharmacists on patients with CRF have showed that pharmacists’ interventions significantly improved the management of anemia, blood pressure, and lipids, as well as calcium and phosphate parameters in these patients, highlighting the crucial role of pharmacists in the health management of CRF patients [12,13,14].

Pharmacy practice in Saudi Arabia is rapidly developing and presents the best and most advanced practices in the region, with multiple practice directions in hospital and non-hospital settings. Additionally, a previous study suggested that apart from dispensing prescribed medication to patients, hospital pharmacists are often involved in unit dose dispensing, the preparation and dispensing of chemotherapies, and IV preparations, including parenteral nutrition [15]. Hospital pharmacists are responsible for medication management and supplies, as well as providing drug information to other health care providers, training residents, and students [15]. Previous literature has highlighted the role of pharmacists in the management of CRF, where pharmacist are able to help slow the progression toward dialysis in CRF patients [9,10,16]. However, limited studies are available that assess how a pharmacist’s current position impacts practices and knowledge, specifically those working as an outpatient hospital pharmacist. The present study aims to assess both the practices and knowledge of pharmacists with respect to the prescribed medications for dialysis patients, taking into consideration the current pharmacist’s position and the quality of counseling provided for dialysis patients.

## 2. Subjects and Methods

### 2.1. Study Design and Pharmacist

This cross-sectional study is based on a survey that was distributed to all outpatient hospital pharmacists, pharmacy practice residents, and discharge-counseling pharmacists during 2015 over a period of 4 months from July to October 2015 at King Abdul-Aziz Medical City-Central Region through a convenience sampling technique. 

### 2.2. Ethical Approval

Institutional review board (IRB) approval was obtained from King Abdullah International Medical Research Center (KAIMRC) in August 2015 with the following reference number (IRBC-580/15-SP-15/150).

### 2.3. Study Questionnaire 

The questionnaire consisted of three major sections. The first section included demographic information (i.e., gender, marital status, career, educational level, degree, current position, years of experience, and source of knowledge). The second section assessed each pharmacist’s practices regarding medication appropriateness review, the dispensing of medication to a caregiver, and the type of drug information resources the pharmacist preferred to use before counseling and dispensing. The third section measured each pharmacist’s knowledge about the most common medications prescribed to dialysis patients and their renal dose adjustments. The answers to questions from the third section can easily be found in databases available at the hospital, including Up-to-date and Micromedex. Verbal consent was obtained from all pharmacists. Confidentiality of data was maintained throughout the study period and thereafter. The reliability of the questionnaire was confirmed with Cronbach’s alpha of 0.72 using results of a pilot study on five pharmacists.

### 2.4. Data Analysis

Descriptive statistics were used to show respondents’ demographic characteristics and list of medications classes. Categorical variables were presented as frequency and percentages. Non-parametric variables were assessed using Chi-square test. A significance level of 0.05 was used for group comparisons.

## 3. Results 

A total of 85 pharmacists received the survey, and 66 completed the questionnaire and were included in the final analysis. Among the pharmacists, 25 (37.9%) were males, 40 (60.6%) were aged between 25–30 years, and slightly more than half were holding a bachelor’s degree in pharmacy, as shown in Table 1. The majority of pharmacists (68%) were working in an outpatients setting, while 13.6% were working as part of a discharge counselling team and 18.2% belonged to a residency program.

Approximately 47% of the surveyed outpatient hospital pharmacists had 1 to 3 years of experience, whereas 56% of the discharge counselling pharmacist had 1 to 4 years of experience. Around 42% of the pharmacists were counselling dialysis patients once a week, while one-third of them (30%) were counselling dialysis patients every two to three days.

The confidence level of pharmacists when counselling hemodialysis patients about their prescribed medications was assessed, and it was found that around two-thirds of them (63.6%) were completely confident, while 32% and only 4.5% were moderately and not confident, respectively.

Pharmacists’ practices regarding the use of drug information resources before counselling was assessed, and results are shown in Table 2. Approximately 34 of the outpatient hospital pharmacists, 7 of the discharge counselling pharmacists, and 6 of the pharmacy practice residents reported that they only refer to drug information resources for new medications in order to review the counselling tips before dispensing them to dialysis patients. Regarding the utilized drug references, Micromedex (outpatient hospital pharmacist (*n* = 40; 88.9%); discharge-counselling pharmacist (*n* = 9; 100%); and pharmacy practice residents (*n* = 11; 91.7%)) and Up-to-date (outpatient hospital pharmacist (*n* = 43; 95.6%); discharge-counselling pharmacist (*n* = 8; 88.9%); and pharmacy practice residents (*n* = 12; 100%)) were the drug information resources most commonly used to answer other drug information questions from dialysis patients.

Practices of the pharmacists based on their current place of working are shown in Table 3. More than half of the outpatient hospital pharmacists 53.3% (*n* = 24) reported that they encourage hemodialysis patients to ask questions about their medications, whereas all of the participating discharge-counselling pharmacists and the majority of pharmacy practice residents 10 (83.3%) reported that they encouraged hemodialysis patients to ask about their medications. All of the participating pharmacists checked each patient’s allergic status before dispensing hemodialysis medications. However, 75.6% of the outpatient hospital pharmacists, 100% of the discharge-counselling pharmacists, and 83.3% of the pharmacy practice residents checked each patient’s medications history before dispensing hemodialysis medications. In regard to the time taken to counsel hemodialysis patients on erythropoietin, 42.2% of the outpatient hospital pharmacists and 22.2% of the discharge-counselling pharmacists took one minute or less to counsel the patients.

There were statistically significant differences between the practices of the three types of pharmacists in terms of checking each patients’ laboratory results before dispensing their medications (*p* = 0.001) and in the length of the counselling sessions (*p* = 0.001). However, statistically significant differences were not found in the other types of practices. The majority of the pharmacy practice residents (75%) and discharge-counselling pharmacists (55.6%) knew that the preferred route for epoetin alfa for hemodialysis patients is IV, while only 40% of the outpatient hospital pharmacists answered this question correctly (*p* = 0.032). Moreover, 83.3% of the pharmacy practice residents knew that there is no need to check the creatinine clearance for chronic dialysis-dependent patients before dispensing their medications. On the other hand, only 20% of the outpatient hospital pharmacists and 22.2% of the discharge-counselling pharmacists answered this question correctly (*p* = 0.001). Other results showed no statistically significant differences in the practices between the three types of pharmacists.

The majority of the outpatient hospital pharmacists, discharge-counselling pharmacists, and the pharmacy practice residents (35 (77.8%), 8 (88.9%); 11 (91.7%), respectively) agreed that oral ciprofloxacin should be given after dialysis session on the same day of dialysis, while 18 (40%), 5 (55.6%), and 9 (75%) of the outpatient, discharge pharmacist and pharmacy practice residents agreed that IV route is preferred for hemodialysis patients to administer epoetin alfa, respectively. Additionally, 91.1% of the outpatient hospital pharmacists, 77.8% of the discharge-counselling pharmacists, and 83.3% of the pharmacy practice residents agreed that the maximum meloxicam dose for dialysis patients was 7.5 mgs per days (Table 4).

## 4. Discussion 

This is the first study focused on assessing practices and knowledge among pharmacists regarding medications prescribed to dialysis-dependent patients in Saudi Arabia. The findings of this study highlight areas that need additional attention in order to improve the quality of care for patients with CRF. Hospital pharmacists were sufficiently aware of routinely dispensed prescriptions from nephrologists. For instance, most pharmacists were aware of the need to check the dosage of the prescribed medication(s), as harmful consequences resulting from adverse drug events could potentially be caused by inappropriate dosages in CRF patients, as shown previously [17,18,19]. The findings of this study are similar to another study carried out among community and hospital pharmacists, which demonstrated that most pharmacists checked for appropriate dosages of renal-excreted drugs [19]. Moreover, in this study, most pharmacists indicated that they followed appropriate practices during the dispensing process and checked the medication history of patients, encouraged patients to ask about their medications, checked patient’s laboratory results, and checked the appropriateness of the timing of medications administration.

Many published studies provide evidence that the counseling patients receive from pharmacists improves medication safety and adherence [20,21]. The finding of this study showed that most pharmacists provided counseling to hemodialysis patients at least once per week. Moreover, 63.6% of pharmacists showed sufficient confidence levels when counseling these patients regarding their medications. This aligns with a previous study conducted in Canada among community pharmacists, which reported that 90% of pharmacists showed confidence when providing counseling to CRF patients about their medications and complications of CRF [22].

In this study, most pharmacists used the Up-to-date followed by the Micromedex database as the main sources for drug information. Use of these databases is the most common practice for obtaining information about medication dosages, precautions, monitoring of parameters, and adverse events. Moreover, these results are in agreement with a previous review reporting that hospital pharmacists are more likely to utilize Up-to-date than community pharmacists, likely because community pharmacists do not have access to Up-to-date [23]. It is important for pharmacists to know the maximum dose for medications, prohibited medication(s), and the appropriate timing to administer certain medications to patients on dialysis, as this could impact a drug’s efficacy. However, 71.2% of hospital pharmacists only sought drug information resources for new medications. Although the answer to some questions in the survey could easily be found in any database, some of the pharmacists did not know that patients should take ciprofloxacin after each dialysis session, that the maximum dose of meloxicam is 7.5 mg, that Augmentin SR should not be prescribed to dialysis patients, or that IV administration is the preferred route for hemodialysis patients [23].

A continuing education (CE) program is crucial to provide pharmacists with sufficient knowledge and skills regarding CRF. This study suggests that a pharmacist’s place of work, whether in an outpatient hospital pharmacy or as part of a discharge-counseling team or residency program, minimally impacts the knowledge and practices of hospital pharmacists. In fact, the only important difference in terms of practices involved the discharge-counseling team and residents, who had better practices toward checking the laboratory results of dialysis patients than outpatient hospital pharmacists. Further, the only important difference in terms of knowledge involved the discharge-counseling team and residents, who had better knowledge regarding the preferred route of administration for epoetin alfa than outpatient hospital pharmacists. The discharge-counseling teams were more knowledgeable about the need to check creatinine clearance in chronic dialysis-dependent patients before dispensing their medication than residents and outpatient hospital pharmacists. Despite these differences, this highlights the great value and outcomes from integrating pharmacists in the medical teams focused on providing care to CRF patients.

There are several limitations to this study. Firstly, the study was conducted on hospital pharmacists working at one hospital. Therefore, these data are not necessarily representative of hospital pharmacists working at other hospitals in Saudi Arabia; however, the concept of the study can be generalized. Second, this study could not assess whether pharmacist looked for the correct answer to the knowledge-based questions in the survey before submitting their answer.

## 5. Conclusions

This study found that most hospital pharmacists present good practices toward dialysis patients regarding their prescribed medications. However, in general, hospital pharmacists have inadequate knowledge regarding certain medications prescribed for hemodialysis patients. Therefore, regular continuing education programs and the integration of knowledge and expertise among pharmacists are essential to provide appropriate patient counseling and consistent optimal therapeutic outcomes. Current place of work and years of experience as a hospital pharmacist were observed to minimally impact practices and knowledge regarding the prescribed medication of dialysis-dependent patients.

## Figures and Tables

**Table 1 healthcare-09-01098-t001:** Characteristics of the hospital pharmacists.

Parameters	Number (*n* = 66)	Percentage (%)
Gender		
Male	25	37.9
Female	41	62.1
Age (years)		
Less than 25	6	9.1
25 to 30	40	60.6
31 to 40	17	25.8
More than 40	3	4.5
Educational level		
Bachelor’s degree in pharmacy science	36	54.5
Pharm D	11	16.7
Master’s degree	11	16.7
PhD	0	0
Residency	8	12.1
Experience		
Outpatient hospital pharmacist		
Less than 1 year	6	13.3
1 year–up to 3 years	21	46.7
4 years–up to 10 years	12	26.7
More than 10 years	6	13.3
Discharge-counselling pharmacist		
Less than 1 year	0	0
1 year–up to 2 years	2	22.2
3 years–up to 4 years	3	33.3
More than 4 years	4	44.4
Pharmacy practice resident		
Less than 1 year	2	16.7
1 year	2	16.7
2 years	5	41.7
More than 2 years	3	25
Frequency of counselling dialysis patients		
Daily	10	15.2
Every two to three days	20	30.3
Once a week	28	42.4
Never	8	12.1

**Table 2 healthcare-09-01098-t002:** Practices of pharmacists towards drug information resources.

How Often You Seek Drug Information Resources to Review the Counselling Tips for Any Medication before Dispensing It to Dialysis Patients Regarding Their Prescribed Medication(s)			
**Variables OP Pharmacist DC Pharmacist PP Resident ***			
Usually for all meds	2 (4.4)	0 (0)	1 (8.3)
Usually for new meds	34 (75.6)	7 (77.8)	6 (50)
Sometimes for all meds	11 (24.4)	5 (55.6)	3 (25)
Sometimes for new meds	8 (17.8)	2 (22.2)	2 (16.7)
Never	0 (0)	0 (0)	0 (0)
**Your usual drug information source to evaluate the prescription is/are**			
Senior pharmacist on duty	22 (48.9)	2 (22.2)	2 (16.7)
Clinical pharmacist	14 (31.1)	6 (66.7)	7 (58.3)
Micromedex	40 (88.9)	9 (100)	11 (91.7)
Up-to-date	43 (95.6)	8 (88.9)	12 (100)
PubMed Database	7 (15.6)	2 (22.2)	3 (25)
Original Guidelines	4 (8.9)	1 (11.1)	3 (25)
Tertiary textbooks	1 (2.22)	2 (22.2)	0 (0)
DI	21 (46.7)	3 (33.3)	2 (16.7)
Other	1 (2.22)	1 (11.1)	0 (0)

* missing data; OP Pharmacist: outpatient hospital pharmacist; DC Pharmacist: discharge-counselling pharmacist; PP Resident: pharmacy practice resident, DI drug information.

**Table 3 healthcare-09-01098-t003:** Practices of the pharmacist before dispensing medications to hemodialysis patients.

Variables	OP Pharmacist	DC Pharmacist	PP Resident
I encourage hemodialysis patients to ask about their medication			
Usually	24 (53.3)	9 (100)	10 (83.3)
Sometimes	20 (44.4)	0 (0)	2 (16.7)
Never	1 (2.2)	0 (0)	0 (0)
I check the patient’s allergy status before I dispense the medication			
Usually	45 (100)	9 (100)	12 (100)
Sometimes	0 (0)	0 (0)	0 (0)
Never	0 (0)	0 (0)	0 (0)
I check the patient’s medication history before I dispense the medication			
Usually	34 (75.6)	9 (100)	10 (83.3)
Sometimes	11 (24.4)	0 (0)	2 (16.7)
Never	0 (0)	0 (0)	0 (0)
I check the patient’s laboratory result before I dispense the medication *			
Usually	11 (24.4)	6 (66.7)	11 (91.7)
Sometimes	25 (55.6)	3 (33.3)	0 (0)
Never	9 (20)	0 (0)	1 (8.3)
I check the appropriateness of the dose of medication			
Usually	40 (88.9)	8 (88.9)	8 (88.9)
Sometimes	5 (11.1)	1 (11.1)	1 (11.1)
Never	0 (0)	0 (0)	0 (0)
I review the appropriateness of the timing of medication administration			
Usually	17 (37.8)	8 (88.9)	7 (58.3)
Sometimes	25 (55.6)	1 (11.1)	5 (41.7)
Never	3 (6.7)	0 (0)	0 (0)
I call the prescribing nephrologist to clarify the prescription			
Usually	28 (62.2)	6 (66.7)	6 (50)
Sometimes	17 (37.8)	3 (33.3)	6 (50)
Never	0 (0)	0 (0)	0 (0)
I take time to counsel hemodialysis patients regarding erythropoietin			
Usually	19 (42.2)	2 (22.2)	2 (16.7)
Sometimes	26 (57.8)	2 (22.2)	10 (83.3)
Never	0 (0)	5 (55.6)	0 (0)
How long does it take you to counsel hemodialysis patients regarding erythropoietin? *			
One minute or less	19 (42.2)	2 (22.2)	2 (16.7)
Two to five minutes	26 (57.8)	2 (22.2)	10 (83.3)
Ten minutes or more	0 (0)	5 (55.6)	0 (0)

* There is a significant association between the variables and the pharmacist’s place of work (*p* < 0.05); OP Pharmacist: outpatient hospital pharmacist; DC Pharmacist: discharge-counselling pharmacist; PP Resident: pharmacy practice resident.

**Table 4 healthcare-09-01098-t004:** Knowledge of the pharmacists towards prescribed medications for dialysis.

Variables	OP Pharmacist	DC Pharmacist	PP Resident
Oral ciprofloxacin should be administering after dialysis session on dialysis days			
Agree	35 (77.8)	8 (88.9)	11 (91.7)
Disagree	4 (8.9)	0 (0)	1 (8.3)
Don’t know	6 (13.3)	1 (11.1)	0 (0)
Using IV route for epoetin alfa is preferred for hemodialysis patients *			
Agree	18 (40)	5 (55.6)	9 (75)
Disagree	8 (17.8)	3 (33.3)	3 (25)
Don’t know	19 (42.2)	1 (11.1)	0 (0)
The pharmacist should check creatinine clearance for chronic dialysis-dependent patients before dispensing their medications *			
Agree	36 (80)	7 (77.8)	1 (8.3)
Disagree	9 (20)	2 (22.2)	10 (83.3)
Don’t know	0 (0)	0 (0)	1 (8.3)
Augmentin SR can be prescribed for dialysis-dependent patients *			
Agree	8 (17.8)	0 (0)	2 (16.7)
Disagree	28 (62.2)	9 (100)	8 (17.8)
Don’t know	9 (20)	0 (0)	2 (16.7)
Darbepoetin alfa should be used for the reduction or control of serum phosphorous in dialysis-dependent patients *			
Agree	5 (11.1)	0 (0)	0 (0)
Disagree	25 (55.6)	8 (88.9)	11 (91.7)
Don’t know	15 (33.3)	1 (11.1)	1 (8.3)
The maximum meloxicam dose for dialysis-dependent patients is 7.5 mgs per day			
Agree	41 (91.1)	7 (77.8)	10 (83.3)
Disagree	2 (4.4)	1 (11.1)	0 (0)
Don’t know	2 (4.4)	1 (11.1)	2 (16.7)

* Significant association between a pharmacist’s place of work and variables (*p* < 0.05).

## Data Availability

Availability of data and materials: The datasets used and analyzed during the current study are available from the corresponding author on reasonable request.

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
