# Peer review of "Assessment of Pharmacists’ Knowledge and Practices towards Prescribed Medications for Dialysis Patients at a Tertiary Hospital in Riyadh Saudi Arabia"

_healthcare, 2021, doi:10.3390/healthcare9091098_

Round 1
Reviewer 1 Report
The introduction is much easier to follow, and the description of pharmacy practice in Saudi Arabia really sets up the paper nicely. This was a great addition for readers who are not as familiar with the standard of care there. methods section has been greatly improved, and it is much easier to follow now. Thank you for your hard work!
I feel the adjustments to this manuscript have fulfilled many of the concerns with the initial draft. This is a helpful resource now. Perhaps it would be worth mentioning that it would be beneficial in future practice to have pharmacists integrated into medical teams focused on CRF?
Author Response
Reviwer-1
The introduction is much easier to follow, and the description of pharmacy practice in Saudi Arabia really sets up the paper nicely. This was a great addition for readers who are not as familiar with the standard of care there. methods section has been greatly improved, and it is much easier to follow now. Thank you for your hard work!
Comment 1: I feel the adjustments to this manuscript have fulfilled many of the concerns with the initial draft. This is a helpful resource now. Perhaps it would be worth mentioning that it would be beneficial in future practice to have pharmacists integrated into medical teams focused on CRF?
Response 1: We indeed appreciate your comment. We totally agree that pharmacists should be integrated in the medical teams taking care of CRF patients. This has been highlighted in the discussion.
Reviewer 2 Report
A properly edited and reviewed manuscript needs to be sent for review. There are red markings (highlights) throughout the manuscript, making it hard to read through. All the references need to be described in the same format in accordance with the journal policy.
What's the objective of the study? the parameters need to be really described, at the same time the importance of these parameters needs to be established. What happens when the pharmacists fail short of these parameters? This study appears to just present some data rather than a critical analysis of what parameters the pharmacists should be graded and what's the current status and implications are.
Mainly its mentioned that the data was collected in 2015, in these world in 5 years things are bound to change drastically in 5 years, especially after the pandemic and this data will not be relevant.
Author Response
Reviwer-2
Comment 1: A properly edited and reviewed manuscript needs to be sent for review. There are red markings (highlights) throughout the manuscript, making it hard to read through. All the references need to be described in the same format in accordance with the journal policy.
Response 1: All have been modified as requested.
Comment 2: What's the objective of the study? the parameters need to be really described, at the same time the importance of these parameters needs to be established. What happens when the pharmacists fail short of these parameters? This study appears to just present some data rather than a critical analysis of what parameters the pharmacists should be graded and what's the current status and implications are.
Response 2: We apologize for the confusion. The present study aims to assess both practices and knowledge of pharmacists with respect to the prescribed medications for dialysis patients taking into consideration the current pharmacist’s position, years of experience and the quality of counseling provided for dialysis patients. Our study revealed the chances of integrating pharmacists in the CRF patients care as well as the areas where pharmacists fall short in, which can be strengthened by providing more training through CEs hours and workshops focused on this issue.
Comment 3: Mainly its mentioned that the data was collected in 2015, in this world in 5 years’ things are bound to change drastically in 5 years, especially after the pandemic and this data will not be relevant.
Response 3: We absolutely agree with the comment, yet the goal of the study is to expose the areas that can be advanced in the pharmacy practice in order to improve the patients care especially when it is related to the CRF patients.
Reviewer 3 Report
This manuscript intends to assess the Pharmacists’ Knowledge and Practices Towards Prescribed Medications for Dialysis Patients at a Tertiary Hospital in Riyadh Saudi Arabia. The topic is relevant for clinical care and the study design and the study procedure are very clear. However, the text needs to be worked, because some phrases are very confusing and the study has a very small sample, so the authors must increase the sample for a correctly interpreting and concluding about the results.
I would like to suggest several paragraphs for revision the text (because the ideas are not clear and confuse):
Intoduction
Last paragraph:” Hospital pharmacist were involved in nontraditional activities such as regulatory activities in specialized areas, such as sterile preparation of medications or chemotherapy and parenteral nutrition”
Results:
Page 5: “Majority of the outpatient hospital pharmacists [35 (77.8%)], the Discharge Pharmacists [8 (88.9%)] and the pharmacy practise residents [11 (91.7%)] agreed that oral ciprofloxacin should be given after dialysis session on the same dialysis days, while 18(40%), 5 (55.6%) and 9 (75%) of the outpatient, discharge pharmacist and pharmacy practise Resi-dents agreed that IV route is preferred for haemodialysis patients to administer Epoetin Alfa, respectively. Also, majority of the participated pharmacists agreed that Maximum Meloxicam dose for dialysis patients was 7.5 mgs per days (Table 4).”
Discussion:
Page 7: “The findings of this study are similar to another study carried out among community and hospital pharmacists, which demon-strated that most pharmacists checked for appropriate dosages of renal-excreted drugs [19]. Moreover, most pharmacists had appropriate practices during the dispensing pro-cess and checked the medication history of patients, encouraged patients to ask about their medications, checked patient’s laboratory results, and checked appropriateness of the time of medications administration”
Page 7- “In this study, most pharmacists used the Up-to-date database followed by the Mi-cromedex database then clinical pharmacists as the main sources for drug information”
Author Response
Reviwer-3
Comment 1: This manuscript intends to assess the Pharmacists’ Knowledge and Practices Towards Prescribed Medications for Dialysis Patients at a Tertiary Hospital in Riyadh Saudi Arabia. The topic is relevant for clinical care and the study design and the study procedure are very clear. However, the text needs to be worked, because some phrases are very confusing and the study has a very small sample, so the authors must increase the sample for a correctly interpreting and concluding about the results.
Response 1: Thank you for your valuable comments and suggestion. This is a single center study that was conducted at a time where we found very limited number of dialysis patients, who visited the study settings. Despite that, we included all the pharmacist who counseled the dialysis patients. Because of the nature of study and hospital setting, we think the sample size is justifiable.
Comment 2: I would like to suggest several paragraphs for revision the text (because the ideas are not clear and confuse):
- Introduction
Last paragraph:” Hospital pharmacist were involved in nontraditional activities such as regulatory activities in specialized areas, such as sterile preparation of medications or chemotherapy and parenteral nutrition”
Response 2a: We appreciate your suggestion. We have corrected the paragraph for clear understanding as follows “Additionally, earlier study suggested that apart from dispensing prescribed medication to the patients, hospital pharmacists were involved in unit dose dispensing, preparation and dispensing of chemotherapies and IV preparations including parenteral nutrition”
- Results
Page 5: “Majority of the outpatient hospital pharmacists [35 (77.8%)], the Discharge Pharmacists [8 (88.9%)] and the pharmacy practice residents [11 (91.7%)] agreed that oral ciprofloxacin should be given after dialysis session on the same dialysis days, while 18(40%), 5 (55.6%) and 9 (75%) of the outpatient, discharge pharmacist and pharmacy practice Residents agreed that IV route is preferred for hemodialysis patients to administer Epoetin Alfa, respectively. Also, majority of the participated pharmacists agreed that Maximum Meloxicam dose for dialysis patients was 7.5 mgs per days (Table 4).”
Response 2b: We apologize for the confusion. This has been corrected in the manuscript for the better understanding , as follows “Majority of the outpatient hospital pharmacists, discharge Pharmacists and the pharmacy practise residents (35 (77.8%)], 8 (88.9%); 11 (91.7%)) agreed that oral ciprofloxacin should be given after dialysis session on the same day of dialysis, while 18(40%), 5 (55.6%) and 9 (75%) of the outpatient, discharge pharmacist and pharmacy practise Residents agreed that IV route is preferred for haemodialysis patients to administer Epoetin Alfa, respectively. Also, 91.1% of outpatient hospital pharmacists, 77.8% of the discharge Pharmacists and 83.3% of the pharmacy practise residents agreed that Maximum Meloxicam dose for dialysis patients was 7.5 mgs per days (Table 4)”
- Discussion
Page 7: “The findings of this study are similar to another study carried out among community and hospital pharmacists, which demon-started that most pharmacists checked for appropriate dosages of renal-excreted drugs [19]. Moreover, most pharmacists had appropriate practices during the dispensing process and checked the medication history of patients, encouraged patients to ask about their medications, checked patient’s laboratory results, and checked appropriateness of the time of medications administration”.
Response 2c: We apologize for the confusion. This has been modified in the text to be better understood.
Page 7- “In this study, most pharmacists used the Up-to-date database followed by the Micromedex database then clinical pharmacists as the main sources for drug information”
Response 2d: We apologize for the confusion. This has been modified in the text to be better understood.
Round 2
Reviewer 3 Report
Although the study has a very small sample, the authors clearly improved the manuscript, therefore, in my opinion, it is now susceptible for publication
This manuscript is a resubmission of an earlier submission. The following is a list of the peer review reports and author responses from that submission.
Round 1
Reviewer 1 Report
Thank you for your work in this area. Pharmacists have such an important role in treating patients with CRF. I would have liked to see a survey focused on outcomes and interventions regarding pharmacist involvement in CRF. I do believe pharmacist involvement in this area has been proven to be effective and the literature highlights successful practice models. I think it would be beneficial to delineate best practices with regards to pharmacist management of CRF.
I found parts of this paper confusing, as outpatient and hospital pharmacists were used interchangeably, but it appears only the outpatient hospital pharmacists were included in the study. Are there additional pharmacists at the hospital who have a strictly inpatient role and were not included in the study? I think it would be helpful to describe the roles of the pharmacists at the practice site in slightly more detail.
Introduction: I found these sentences to be confusing: "However, limited studies are available to assess how a pharmacist’s current position impacts attitude and knowledge, specifically those working as an outpatient pharmacist. The present study assessed the quality of hospital pharmacist counseling by assessing both attitudes and knowledge of pharmacists regarding prescribed medications for dialysis patients" - It is mentioned there is limited information, specifically related to outpatient pharmacists, but the next sentence then says this focuses on hospital pharmacists. I think the study actually focused on outpatient hospital pharmacists. I think it is best to try to have more of a flow to these two sentences, as to some readers, they do not flow (as the first sentence says not enough data with outpatient, and the second sentence disregards that sentiment as it states the current study is focused on hospital pharmacists).
I would make the introduction more robust and pull in some of these studies were pharmacists have an impact and highlight specific examples of interventions.
It may be helpful to describe the scope of pharmacy practice in Saudi Arabia, as other countries have pharmacists embedded as part of the renal team. Is this relatively new to have pharmacists engaged in CRF care? I think it would give more context as to why your survey is fulfilling a gap in the literature. I am not sure, with how the paper is written now, that it fulfills a significant gap.
Methods: What drug information sources did the pharmacists have access to? Was it just Up to Date and Micromedex? Can you describe your enrollment process? The abstract describes the technique used to survey pharmacists, but this needs to be described in the actual text of the paper as well.
Results: I am not sure "attitude" is the right way to describe the part of the survey that deal with the actions of the pharmacists. I think it is better described as "actions of the pharmacists" or "practices of the pharmacists". I would consider "attitude" to be "I believe it is important to check the patient's allergy status..."
Discussion: you did a nice job comparing this to other literature and citing it is the first study of its kind in Saudi Arabia. What are you hoping to do with these results? How do these results have a significant impact on practice? With the audience of this journal, how can these results help those in other countries (given the international audience)?
Reviewer 2 Report
The manuscript covers an important topic. Revisions are needed to meet publications standards:
- Grammar must be improved throughout the whole paper.
- Terminology must be corrected. For example "convenient sampling" should be "convenience sampling" with appropriate referencing to this method.
- There are two Table 3's in the manuscript.
- Attitude items are actually questions about respondents' behaviors and should be corrected to be clearer to readers.
- Some items do not have response categories that make any sense.
- It appears that items in Table 4 have correct answers. These should be described and then % correct, % incorrect and % not sure should be reported.
- The case is made for continuing education in this area. Linking the rest of the paper with this recommendation could work well. If you have any questions related to the need for continuing education or any comments about this, that would strengthen the paper.